# A BET Protein Inhibitor Targeting Mononuclear Myeloid Cells Affects Specific Inflammatory Mediators and Pathways in Crohn’s Disease

**DOI:** 10.3390/cells11182846

**Published:** 2022-09-12

**Authors:** Ahmed M. I. Elfiky, Ishtu L. Hageman, Marte A. J. Becker, Jan Verhoeff, Andrew Y. F. Li Yim, Vincent W. Joustra, Lieven Mulders, Ivan Fung, Inmaculada Rioja, Rab K. Prinjha, Nicholas N. Smithers, Rebecca C. Furze, Palwinder K. Mander, Matthew J. Bell, Christianne J. Buskens, Geert R. D’Haens, Manon E. Wildenberg, Wouter J. de Jonge

**Affiliations:** 1Tytgat Institute for Liver and Intestinal and Research, Amsterdam Gastroenterology & Metabolism, Amsterdam University Medical Centers, University of Amsterdam, 1105 BK Amsterdam, The Netherlands; 2Immunology Research Unit, GSK Medicines Research Centre, Stevenage SG1 2FX, UK; 3Department of Gastroenterology and Hepatology, Amsterdam University Medical Centers, Amsterdam Gastroenterology Endocrinology Metabolism (AGEM), University of Amsterdam, 1105 AZ Amsterdam, The Netherlands; 4Department of Molecular Cell Biology & Immunology, Amsterdam Infection & Immunity Institute and Cancer Center Amsterdam, Amsterdam University Medical Centers, Free University Amsterdam, 1081 HV Amsterdam, The Netherlands; 5Genome Diagnostics Laboratory, Department of Clinical Genetics, Amsterdam Reproduction & Development, Amsterdam University Medical Centers, University of Amsterdam, 1105 AZ Amsterdam, The Netherlands; 6Department of Surgery, Amsterdam UMC, University of Amsterdam, 1081 HV Amsterdam, The Netherlands; 7Department of Surgery, University of Bonn, 53127 Bonn, Germany

**Keywords:** BET inhibitor, CES1, IBD

## Abstract

Background: Myeloid cells are critical determinants of the sustained inflammation in Crohn’s Disease (CD). Targeting such cells may be an effective therapeutic approach for refractory CD patients. Bromodomain and extra-terminal domain protein inhibitors (iBET) are potent anti-inflammatory agents; however, they also possess wide-ranging toxicities. In the current study, we make use of a BET inhibitor containing an esterase sensitive motif (ESM-iBET), which is cleaved by carboxylesterase-1 (CES1), a highly expressed esterase in mononuclear myeloid cells. Methods: We profiled CES1 protein expression in the intestinal biopsies, peripheral blood, and CD fistula tract (fCD) cells of CD patients using mass cytometry. The anti-inflammatory effect of ESM-iBET or its control (iBET) were evaluated in healthy donor CD14^+^ monocytes and fCD cells, using cytometric beads assay or RNA-sequencing. Results: CES1 was specifically expressed in monocyte, macrophage, and dendritic cell populations in the intestinal tissue, peripheral blood, and fCD cells of CD patients. ESM-iBET inhibited IL1β, IL6, and TNFα secretion from healthy donor CD14^+^ monocytes and fCD immune cells, with 10- to 26-fold more potency over iBET in isolated CD14^+^ monocytes. Transcriptomic analysis revealed that ESM-iBET inhibited multiple inflammatory pathways, including TNF, JAK-STAT, NF-kB, NOD2, and AKT signaling, with superior potency over iBET. Conclusions: We demonstrate specific CES1 expression in mononuclear myeloid cell subsets in peripheral blood and inflamed tissues of CD patients. We report that low dose ESM-iBET accumulates in CES1-expressing cells and exerts robust anti-inflammatory effects, which could be beneficial in refractory CD patients.

## 1. Introduction

Crohn’s disease (CD) is a complex immune-mediated disease presenting as chronic inflammation of the gastrointestinal tract [1]. Immunomodulatory therapies are the mainstay of treatment, which include steroids, thiopurines, and biological agents, such as anti-TNF (infliximab, adalimumab), anti-α4β7 integrin (vedolizumab), and anti-IL12p40 (ustekinumab) [2,3]. Current therapeutic strategies have a response rate of approximately 30% [4], making non-responsiveness to therapy, along with disease progression to a severe clinical phenotype, such as fistulizing CD (fCD) [3,5,6], a clinical challenge. Unfortunately, surgical removal of affected intestinal tissue is required in approximately 70% of CD patients [7], highlighting the unmet need to introduce new treatments that are better tolerated and demonstrate superior clinical efficacy. 

Bromodomain and extra terminal (BET) domain-containing proteins are a family of epigenetic readers (BRD2, BRD3, BRD4, and BRDT) that bind acetylated lysine residues of histone proteins to allow for transcriptional complex formation and gene expression (8). Regarding regulation of the immune response, the BET proteins are essential for the transcription of several inflammation-related genes and have therefore been targets of interest in drug development for inflammatory diseases and cancer [8,9]. Small molecule inhibitors for BET proteins (iBET) show demonstrable therapeutic benefits in multiple pre-clinical models of inflammatory diseases [10,11]. However, in relation to murine models of IBD, the outcome was uncertain. In T-cell mediated colitis, iBET improved colon inflammation [12], while in a dextran sulfate sodium (DSS)-induced colitis mouse-model (with chemically driven epithelia damage), colon inflammation was aggravated [13]. This unexpected outcome in the DSS-induced colitis model is largely explained by iBET toxicity to colon epithelium [14], which may limit the beneficial immunosuppressive effect. Several iBET compounds have been investigated in randomized clinical trials in cancer patients; overall, the clinical efficacy was limited, despite promising outcomes in pre-clinical cancer models [15]. Multiple adverse events (AE) were reported [16,17,18], including thrombocytopenia, anemia, neutropenia, diarrhea, and pneumonia, which limit further clinical development of these iBET compounds. Redirecting iBET to specific cell types may limit the wide-range toxicity and allow efficacy at a very low dose.

Esterase sensitive motif (ESM) technology has previously been described to achieve cell specific accumulation of the active drug, targeting mononuclear myeloid cells based on the presence of carboxylesterase-1 (CES1) [19]. This approach has demonstrated therapeutic benefits in pre-clinical models of arthritis [19], colitis [20], peritonitis, and atherosclerosis [21], using an ESM-conjugated histone deacetylase enzyme (HDAC) inhibitor. We hypothesize that an ESM-conjugated iBET could improve tolerability by specifically targeting iBET to CES1 expressing cells within CD. This approach might be of great interest in the treatment of CD, as myeloid cells are key players in sustained inflammation [1]. In the current study, we are the first to investigate the efficacy of an iBET with an esterase sensitive motif (GSK3361191 or ESM-iBET) application in CD. We first profile CES1 expression in multiple CD or inflammatory bowel related clinical samples, such as intestinal biopsies, PBMCs from CD patients, and curettage material from fistula tracts of CD patients. Next, we provide comprehensive analysis of proteins and genes modulated by iBET in monocytes and compare the effect of ESM-iBET (GSK3361191) with the non-hydrolysable iBET control (GSK3235220).

## 2. Materials and Methods

Detailed information on the materials, methods, and associated references can be found in the Appendix A (Appendix A).

### 2.1. Compounds

GSK3361191 (ESM-iBET) and its non-hydrolysable control GSK3235220 (iBET) were provided by GlaxoSmithKline (Stevenage, UK). GSK3361191 (ESM-iBET) is a BET inhibitor with an esterase sensitive motif (ESM), and GSK3235220 (iBET) is a pan BET-inhibitor. GSK3361191 (ESM-iBET) is similar in its mechanism of action compared to an earlier published compound, GSK3358699 [22], and is cleaved by carboxylesterase 1 (CES1), which allows selective hydrolyzation within CES1 positive cells to a charged, intracellularly retained drug [19]. For in vitro studies, both compounds were dissolved in 100% DMSO and used at concentration ranges of 0.002 µM to 10 µM.

### 2.2. Human Clinical Samples

The following clinical samples were analyzed: intestinal biopsies of inflammatory bowel disease (IBD) patients (CD patients and ulcerative colitis patients), PBMCs of CD patients, and fistula tract tissue of fCD patients. Samples were obtained from the department of gastroenterology and/or surgery at the Amsterdam UMC, University of Amsterdam, under the approval of the accredited Medical Ethics Committee (METC #NL53989.018.15, #NL75341.018.20) or the biobank committee of the Amsterdam UMC (178 #A201470). Intestinal biopsies, PBMCs, and fistula samples were cryopreserved and handled according to the methodology published by Konnikova et al. [23]. Detailed information about the clinical sampling and experimental details are found in the SM, “Additional information on clinical sampling and experimental work-up.” The patient characteristics can be found in Appendix A.

### 2.3. Mass Cytometry 

Human clinical samples (see description above) were immunophenotyped using a CyTOF Helios mass cytometer. Staining, barcoding, data acquisition, and mass cytometry analysis are described in the supplementary methods. We made use of three different antibody panels: a biopsy panel, a PBMC panel, and a fistula panel (found in Appendix A). Acquisition was performed on the Cytometry by time of flight (CyTOF)3-Helios. The sample was diluted in H_2_O and supplemented with 10% *v/v* of EQ Four Element Calibration beads (Fluidigm, San Francisco, U.S.). After acquisition, the data were normalized, and individual files were deconvoluted using the CyTOF software v6.7 functions. Different lineages (B cells, CD4 T cells, CD8 T cells, myeloid cells, and NK cells) were clustered using FlowSOM and subsequent manual annotation [24]. Data is visualized using tSNE, a dimensionality reduction tool for high-dimensional single-cell data in R [25,26].

### 2.4. Human PBMCs and Monocyte In Vitro Culture

We obtained buffy coats (healthy donors) from Sanquin Blood Bank, Amsterdam, and isolated the peripheral blood mononuclear cells (PBMCs) according to standard Ficoll (GE Healthcare Bio-Sciences AB, Danderyd, Sweden) density gradient centrifugation protocol [27]. We further isolated CD14+ monocytes using a human CD14 positive selection kit (Miltenyi Biotech, Germany). 

For the in vitro culture for cytokine analysis, the CD14+ monocytes or PBMCs were pre-treated for 1 h with a concentration range of 0.002, 0.01, 0.04, 0.156, 0.625, 2.5, and 10 µM of either GSK3361191 (ESM-iBET), non-hydrolysable control GSK3235220 (iBET), or DMSO. After 1 h, cells were washed to remove the extracellular compound, stimulated with 10 ng/mL LPS dissolved in RPMI medium (Thermofisher Scientific, Waltham, MA, USA), and incubated overnight. Supernatant was collected, and cytokine (TNFα, IL1β, IL6) analysis was performed using Cytometric Bead Array (CBA) (BD Biosciences, Australia). Intracellular TNFα protein expression was detected by flow cytometry analysis (FACS Fortessa, BD Biosciences, NJ, USA) and analyzed using FlowJo software (Treestar Inc., Ashland, OR, USA). Cytokine data is visualized by normalizing the actual measured values to the DMSO control to correct for the biological variation in every individual donor. 

For in vitro culture for RNA transcriptomics analysis, CD14+ monocytes were pre-treated for 1 h with 40 nM of either GSK3361191 (ESM-iBET), non-hydrolysable control GSK3235220 (iBET), or DMSO. After 1 h, cells were washed and stimulated with 10 ng/mL LPS dissolved in RPMI medium (Thermofisher Scientific, Waltham, MA, USA) and incubated for 4 h. 

### 2.5. Ex Vivo Derived CD Fistula Tract Cells Culture

CD fistula samples were obtained from fistulizing CD patients undergoing surgery (seton placement/removal, or inspection) at the Amsterdam UMC, location AMC. Fistula scrapings were mechanically digested by mashing and flushing through a 100 µm cell strainer (BD Falcon, Franklin Lakes, NJ, USA) placed on a 50 mL tube (Sarstedt, Germany), and immune cells were isolated using Ficoll isolation [26]. Immune cells were incubated for 16 h with a concentration range of 0.0025, 0.01, 0.04, 0.625, 2.5, and 10 µM of either GSK3361191 (ESM-iBET), non-hydrolysable control GSK3235220 (iBET), or DMSO resolved in RPMI medium (Thermofisher Scientific, Waltham, MA, USA). After incubation, the cells were either collected for cytokine analysis by CBA or flow cytometric analysis of intracellular TNFα. Cytokine data is visualized by normalizing the actual measured values to the DMSO control to correct for the biological variation in every individual patient. 

### 2.6. RNAseq Transcriptome Analysis 

Transcriptomic analyses were performed through RNA sequencing. Briefly, mRNA was isolated using the Isolate RNA mini kit (Bioline, UK) and converted into cDNA. Subsequently, cDNA was sequenced in a 150 bp paired-ended fashion on the Illumina NovaSeq6000 to a depth of 40 million reads at the Amsterdam UMC Genomics Core Facility. The quality control of the reads was performed with FastQC (v0.11.8) and summarization through MultiQC (v1.0) [28]. The raw reads were aligned to the human genome (GRCh38) using STAR (v2.7.0) [29] and annotated using the Ensembl v95 annotation. Post-alignment processing was performed through SAMtools (v1.9), after which the reads were counted using the featureCounts function found in the Subread package (v1.6.3) [30]. Differential expression (DE) analysis was performed using the Bioconductor (v3.14) [31] package DESeq2 (v1.22.2) [32] in the R statistical environment (v3.46.0) [33], where we compared both BET-inhibitors (GSK3361191/ESM-iBET and control GSK3235220/iBET) with DMSO or GSK3361191/ESM-iBET with DMSO. Differentially expressed genes (DEGs) were defined as genes whose difference presented a Benjamini–Hochberg-adjusted *p*-value < 0.05. Gene set enrichment analysis was conducted with the fgsea package (v1.20) [34] using the Kyoto Encyclopedia of Genes and Genomes (KEGG) database as a reference [35]. Visualizations were created in ggplot2 (v3.3.5) [36].

### 2.7. Statistical Analysis 

Statistical analysis was performed using GraphPad Prism 7.0 (GraphPad Software, La Jolla, CA). Statistical testing was performed using a two-way ANOVA test; * *p* ≤ 0.05, ** *p* ≤ 0.01, *** *p* ≤ 0.001, **** *p* ≤ 0.0001; SEM is the standard error of the mean.

## 3. Results

### 3.1. Immuno-Phenotyping of IBD Intestinal Biopsies Reveals Specific CES1 Expression in CD14^+^ Myeloid Cell Population

In order to explore the potential application of ESM-conjugated molecules in the treatment of IBD, we examined CES1 expression in IBD clinical samples using mass cytometry analysis, with particular emphasis on the inflamed local tissue environment. First, we investigated the immune cell composition in intestinal biopsies collected from IBD patients during endoscopy. Biopsies were taken from six IBD patients and collected from inflamed areas (n = 6) or non-inflamed areas (n = 2). We were able to identify CD27− (naïve) and CD27+ (memory) B cells (CD45+CD45RA+HLA-DR+CD69+CD44+), CD4 T cells (CD45+CD3+CD69+ CD2+CD4+CD5+CD28+), CD8 T cells (CD45+CD3+CD8a+CD5+), mononuclear myeloid cells (CD45+CD14+CES1+HLA-DR+CD11a+CD44+CD11b+), epithelial cells (CD45−EpCAM+CD95+), and NK cells (CD45+CD45RA+CD161+CD2+CD7+) (Figure 1A,C). Furthermore, we identified a CD4−CD8−T cell population (CD45+CD3+CD69+CD44+), which is the double negative T cell fraction [37]. The mononuclear myeloid cells demonstrated a high expression of CD14 and a low expression of CD16. Among the defined CD intestinal biopsy-derived cells, we demonstrated that CES1 expression was restricted to the mononuclear myeloid population (Figure 1B,C), with a median of 80% CES1 expressing cells within this population (Figure 1D). Interestingly, we also demonstrated some CES1+ cells within the EpCAM+ fraction (epithelial cells), compared to the B cell, T cell, or NK cell fraction, however, with much less in frequency compared to CD14+ myeloid cells. 

### 3.2. CES1 Is Expressed in Peripheral Blood Mononuclear Myeloid Cells of CD Patients 

Next, we aimed to define CES1-expressing immune cell populations in the peripheral blood of CD patients, determining whether CES1 expression, among identified populations, differs between biological therapy-responsive and non-responsive patients. Therefore, we isolated PBMCs from CD patients treated with the biological agent vedolizumab and analyzed them using CyTOF to define CES1 expression among identified immune cell populations. We identified CD4+ T cells, CD8+ T cells, NK cells, and B cells and extended the analysis of myeloid subsets to include classical monocytes (CD14+++CD16−), intermediate monocytes (CD14++CD16+), non-classical monocytes (CD14+CD16++), cDCs (CD16−CD14−HLADR++CD11a+ CD2+), and pDCs (CD16−CD14−HLADR+++CD45RA+ CD2++) (Figure 2A,C), although in the latter, we are missing the typical CD123 marker for classifying pDCs [38]. We demonstrated a high CES1 expression in the above-mentioned mononuclear myeloid populations (Figure 2B,C), with the highest expression in the intermediate monocytes and the non-classical monocytes (Figure 2D). PBMCs collected from vedolizumab non-responsive patients (n = 6) did not statistically differ from responsive patients in the percentage of CES1+ cells in these myeloid subsets (Figure 2D). 

### 3.3. CES1 Is Expressed within Macrophages and Dendritic Cells Retrieved from Inflamed Fistula Tracts of CD Patients

Next, we explored the immune cell composition within the highly inflamed tissue environment of the fistula tracts of CD patients, using a penetrating phenotype that underwent surgical intervention (n = 13). We analyzed cells retrieved from CD fistula tract curettage material using CyTOF and established the presence of different immune cell subsets, including basophils, B cells, CD4+ T cells, CD8+ T cells, NK cells, and mononuclear myeloid cell subsets (Figure 3A,C). CES1 was highly expressed in the mononuclear myeloid cell compartment that includes macrophages (CD68+CD14+HLA-DR+CD44+CD11b++), CD163+ macrophages (CD68+CD14+HLA-DR++CD44++CD11b+CD163+), DCs (CD11c+HLA-DR+++CD14−CD141−CD123−), CD141+DCs type 1 (CD11c+HLA-DR+++CD14−CD141+CD123−), and pDCs (CD11c−HLA-DR++CD14−CD141−CD123+) (Figure 3B,C). We observed that the percentage of CES1 positive cells was higher in the macrophage populations compared to the DCs populations, with the lowset CES-1 expression noted in pDCs (Figure 3D). Other identified immune cell types showed minimal to no CES1 expression (Figure 3C,D).

### 3.4. ESM-iBET Demonstrated an Increased Anti-Inflammatory Effect Compared to Its Non-Hydrolysable Control iBET in Healthy Donors Monocytes

Next, we evaluated the anti-inflammatory efficacy of an ESM-iBET (GSK33611910) and compared this to its non-hydrolysable iBET control (GSK3235220) in LPS stimulated monocytes and PBMCs from healthy donors. Since ESM-iBET specifically accumulates in CES1-positive myeloid cells, we hypothesized a more potent immunosuppressive effect in the monocytes. PBMCs from healthy donors (n = 3) were treated with a concentration range (0.002–10 µM) of ESM-iBET or iBET, and TNFα expression from CD14-expressing cells was determined using flow cytometry (Figure 4A). The frequency of TNFα-expressing CD14+ monocytes was significantly reduced in ESM-iBET treated cells (23.9%) compared to iBET 50.3%) or DMSO treated cells (49.0%) at 156 nM (Figure 4B).

Next, CD14+ monocytes from healthy donors (n = 3) were treated with a concentration range (0.002–10 µM) of ESM-iBET or iBET, and secreted inflammatory cytokines (IL1β, IL6, TNFα) were quantified with cytometric bead array (CBA). ESM-iBET demonstrated significantly increased potency to inhibit IL1β, IL6, and TNFα secretion when compared to the control iBET, with calculated IC50 of 9.6 nM vs. 257.4 nM (IL1β), 19.1 nM vs. 269.4 nM (IL6), and 14.8 nM vs. 145.6 nM (TNFα) for ESM-iBET or iBET, respectively (Figure 4C). 

### 3.5. ESM-iBET and iBET Similarly Inhibited Inflammatory Cytokine Secretion from CD Fistula Tract-Derived Immune Cells in a Dose-Dependent Fashion

Subsequently, we extended these observations to evaluate ESM-iBET (GSK33611910) or iBET (GSK3235220) anti-inflammatory activity in immune cells retrieved from inflamed fistula tract of CD patients. Ex vivo isolated immune cells were treated with a concentration range (0.002–10 µM) of ESM-iBET or iBET, and inflammatory cytokines (IL1β, IL6, TNFα), secreted overnight, were quantified. Both inhibitors efficiently reduced secreted inflammatory cytokines at relatively higher concentrations when compared to PBMC and monocyte cultures, with calculated IC50 of 441.3 nM vs. 1185 nM (IL1β), 280.3 nM vs. 358.3 nM (IL6), and 150.4 nM vs. 255.2 nM (TNFα) for ESM-iBET and iBET, respectively. There was no significant difference between the concentration of ESM-iBET and iBET required to inhibit inflammatory cytokine secretions from fCD ex vivo isolated immune cells (Figure 4D).

### 3.6. ESM-iBET Influences the Transcription of Inflammatory Related Genes and Pathways with Increased Potency over iBET in Blood CD14^+^ Monocytes

In order to gain more insight into the transcriptome changes mediated by BET inhibition, we compared monocytes pre-treated with ESM-iBET or iBET to the DMSO control treatment. In line with earlier functional assays (Figure 4), we expected the immunosuppression of different inflammatory pathways and aimed to assess the differences between ESM-iBET and iBET. In addition to this, CD14+, monocytes were isolated from healthy donor PBMCs (n = 5), pre-treated with 40 nM of either ESM-iBET (GSK33611910), iBET (GSK3235220), or DMSO, then stimulated with LPS for 4 h. Through principal component (PC) analysis, we observed a separation between DMSO+LPS, ESM-iBET+LPS, and iBET+LPS from non-LPS DSMO in PC2 (Figure 5A), suggesting an association within PC2 with LPS stimulation. By contrast, PC1 presented a separation between ESM-iBET pre-treated samples and the other sampels, which was not immediately visible for iBET, indicating that ESM-iBET pre-treatment affects monocytes more profoundly than iBET.

We identified 253 significantly differentially expressed genes (DEGs), of which 163 were upregulated and 90 were downregulated. Interestingly, visualizing the top 20 upregulated or downregulated DEGs in response to BET inhibition suggested that while both BET inhibitors functioned concordantly, ESM-iBET conferred a stronger effect, as opposed to iBET, at equimolar level concentrations (Figure 5B), in line with earlier cytokine inhibition data on monocytes (Figure 4A–C). Moreover, we compared the effect sizes of the top 10 upregulated or downregulated inflammation-related genes using the Wald statistic when comparing ESM-iBET to DMSO on the x-axis, and iBET compared to DMSO on the y-axis (Figure 5C). While most of the genes were affected in same direction by both BET inhibitors, the strongest effect was observed for ESM-iBET (Figure 5C). Focusing on the inflammation-related genes, we identified chemokines (CCL14—CCL25), cytokines (IL12B—IL36B—CSF2), and members of the MAPK signaling pathway (MAP3K4—MAP3K20) to be the most significantly downregulated after pretreatment with ESM-iBET relative to DMSO. Among the top upregulated ESM-iBET targets, we identified pro-apoptotic genes (BCL2L11), phosphatases such as PHLPP2 and DUSP7, which are known to dephosphorylate effective mediators of the AKT [39] and MAPK [40,41] signaling pathways, respectively, and therefore, act as negative regulators. 

Furthermore, we performed gene set enrichment analysis (GSEA) with functional annotation using the Kyoto Encyclopedia of Genes and Genomes (KEGG) pathways. Multiple inflammation-related pathways were among the top significantly negatively enriched pathways in response to ESM-iBET pre-treatment (Figure 5D). We could identify pathways of therapeutic and pathogenic relevance to CD, including cytokine–cytokine receptor interaction, TNFα signaling, JAK-STAT signaling, NF-kappa B signaling, MAPK signaling, NOD-like receptor signaling, and PI3K-Akt signaling pathways [42,43,44,45,46] (Figure 5D,E).

### 3.7. ESM-iBET Potently Modulates Cytokines/Chemokines Transcription in Monocytes

Next, we aimed to explore the inflammation related cytokines and chemokines ligands and their receptors that are targeted by BET inhibition in the monocytes. The gene set enrichment analysis (GSEA) of the cytokine–cytokine receptor interaction pathway was negatively enriched in response to ESM-iBET treatment when compared to the DMSO-treated monocytes (Figure 5E). Multiple chemokines were significantly inhibited, including CCL1, CCL4, and CXCL5. Notably, CXCL14 expression was found to be enhanced (Appendix A). In regards to the effect on cytokines following ESM-iBET treatment, we identified the inhibition of multiple genes related to the IL6, IL1, IL10, TNFα, TGF-β, and interferon families. Alternatively, the expression of selected cytokine receptors was attenuated, including IL1R1, IL17RA, and IFNGR2 (Appendix A).

### 3.8. ESM-iBET Affects Transcription of Key Pathways in CD Pathogenesis, Such as TNFα, JAK-STAT, NF-kB, and NOD2 Signaling

Next, we explored whether ESM-iBET affects the transcription of inflammatory pathways and target genes of therapeutic interest in CD. TNFα and JAK-STAT signaling pathways are key players in CD pathogenesis and therefore, are of important therapeutic relevance [44]. GSEA showed a strong negative enrichment of both the TNFα (Figure 5E, Appendix A) and JAK-STAT signaling pathways (Figure 5E, Appendix A) in ESM-iBET-treated monocytes compared to the DMSO-treated monocytes. We further explored effector genes and downstream signaling pathways to identify ESM-iBET targets. We showed that TNFα itself was strongly downregulated by ESM-iBET treatment, while both TNFα receptors (TNFR1 and TNFR2) remained unaffected. Additionally, ESM-iBET targeted the inflammatory signaling pathways that are activated farther downstream of TNFα signaling, such as the MAPK, NF-kB, and PI3K-Akt pathways (Figure 5D, Appendix A). Moreover, within the JAK-STAT downstream signaling mediators, we identified STAT5A to be strongly downregulated by ESM-iBET treatment, while SOCS6 and PIAS3, known negative regulators of JAK-STAT signaling [47], were strongly upregulated (Appendix A). Moreover, the NF-κB signaling pathway was found to be negatively enriched in response to ESM-iBET treatment compared to DMSO treatment (Figure 5E), as is illustrated by a downregulation of RELA (P65), a key functional subunit in the NF-κB canonical pathway (Appendix A). Furthermore, GSEA of the NOD2 signaling pathway was negatively enriched by ESM-iBET treatment (Figure 5E), with key effector caspase genes (CASP1, CASP4, and CASP5) being efficiently downregulated (Appendix A).

## 4. Discussion

Dysregulated innate immunity plays a fundamental role in the sustained and recurring inflammation in CD. Through epigenetic mechanisms, BET proteins are essential for inflammatory gene expression [8]. In the current study, we investigated the potential benefits of BET inhibition, specifically in the mononuclear myeloid cell compartment in the context of CD, and highlighted potential mechanisms of the iBET-mediated anti-inflammatory response in monocytes. We introduced a mononuclear myeloid cell-targeted iBET (ESM-iBET GSK3361191) as a small molecule BET inhibitor that is effective in reducing inflammatory cytokine production from mononuclear myeloid cells due to its accumulation in CES1 expressing cells (19). ESM-iBET is expected to be specifically retained in CES1-expressing cells, while demonstrating a reduced effect in non-CES1 expressing cells (19, 22). In order to understand the exact profile of CES1 expression in inflamed and non-inflamed CD tissue, we performed CyTOF-based profiling of CES1 protein expression in CD patient cells, which revealed a specific pattern of expression, confined within mononuclear myeloid cell populations, across peripheral blood cells, in inflamed intestinal tissue, and in fistula tract-derived cells. 

In intestinal biopsies, CES1 was mainly restricted to the CD14-expressing myeloid cells. An earlier study demonstrated a toxic effect of non-selective BET-inhibitors on intestinal epithelial cells (14) and a worsening of DSS-induced colitis in a mouse model upon BET-inhibition (13), suggesting the toxicity of non-selective BET-inhibition on intestinal epithelial cells. In this context, we demonstrated that the majority of EpCAM+ epithelial cells did not express CES1, which is of particular interest here for the application of ESM-iBET in inflammatory intestinal diseases by sparing intestinal epithelial cells, while targeting inflammatory mononuclear myeloid cells. However, a small fraction of CES1-expressing epithelial cells (median of 8.5%) is noted, which may require further investigation to determine whether this reflects a specific epithelial cell subset and whether it may have a detrimental effect on CES1-assisted drug delivery applications in IBD.

In CD peripheral blood PBMCs, CES1 expression is confined to the myeloid compartment, including classical monocytes (CD14+++CD16−), intermediate monocytes (CD14++CD16+), non-classical monocytes (CD14+CD16++), cDCs (CD16−CD14−HLADR++CD11a+CD2+), and pDCs (CD16−CD14−HLADR+++CD45RA+CD2++), similar to the expression pattern demonstrated earlier in healthy donor blood (20). Interestingly, we observe the highest CES1 expression among circulating non-classical (CD14+CD16++) monocytes from CD patients, in contrast to healthy donors, where CES1 expression was the highest among classical monocytes (CD14+++CD16−) (20). This may potentially be reflective of a preferential trafficking of CES1-expressing classical monocytes to the locally inflamed colon in CD patients, as classical monocytes are recruited to fuel the inflammatory process [48,49]. The CES1 expression within mononuclear myeloid cells is also shown within ex vivo cells retrieved from CD fistula tracts, supporting the possibility that these cells can be potentially targeted by ESM-iBET to achieve a therapeutic response in these patients. A high CES1 expression is found across the macrophage subsets within these fistula samples. Moreover, our earlier findings show that CES1-expressing macrophages are enriched in inflamed CD intestinal mucosa (20); altogether, this suggests that these macrophage populations in the CD inflamed environment would be the most targeted cells of ESM-iBET. Notably, no significant difference in CES1 expression is observed between identified macrophage and DCs subsets, except for pDCs, which show relatively lower CES1 expression, possibly reflecting an equally pronounced effect of ESM-iBET on those tissue CES1+ myeloid subsets. 

Using a non-hydrolysable iBET control (GSK3235220), we demonstrated a higher potency of ESM-iBET (GSK3361191) to inhibit inflammatory cytokines in CES1-expressing monocytes in both PBMCs or purified CD14+ monocyte culture compared to equimolar concentrations of iBET (GSK3235220). This validates the specific CES1 assisted delivery of ESM-iBET and the augmented anti-inflammatory effect. However, despite the inhibition of inflammatory cytokine secretion from ex vivo immune cells retrieved from CD fistula tracts by ESM-iBET, no differential efficacy over similar concentrations of iBET was observed. This can be explained by the presence of other immune cells, such as T cells and B cells mixed with a low yield of CES1-expressing myeloid cells, within these cell preparations compared to PBMCs and purified monocytes; therefore, the contribution of ESM-iBET targeted CES1-expressing cells to overall secreted cytokines is minimal to demonstrate an observed difference. 

The transcriptomic analysis of ESM-iBET (GSK3361191)-treated monocytes demonstrates a potent inhibitory effect on multiple inflammation related genes and pathways. This is in line with earlier reports using other iBET in human primary monocytes [50], human and murine microglial cell lines [51,52], and murine bone marrow derived macrophages [53] Additionally, we demonstrated a higher potency of ESM-iBET (GSK3361191) compared to iBET (GSK3235220) control at a low dose of 40 nM. Among the downregulated inflammatory genes, we could identify multiple targets of therapeutic relevance to CD, such as IL12B and TNFα, known therapeutic targets of biological agents such as ustekinumab, infliximab, or adalimumab (2). Additionally, ESM-iBET efficiently downregulated oncostatin M (OSM), IL1α, and IL1R,1 which were previously reported to be involved in anti-TNF therapy non-responsiveness [54,55,56] in IBD. Interestingly, among the key anti-inflammatory mediators, ESM-iBET upregulated the TGFβ receptor. In this context, TGFβ signaling by intestinal DCs [57] and circulating monocytes [58] exerts an anti-inflammatory effect, and therapeutically augmenting this pathway shows therapeutic benefits in CD patient clinical trials [59].

The monocyte transcriptional analysis detailed herein demonstrates that ESM-iBET can efficiently target key components of multiple inflammatory pathways involved in the pathophysiology of CD. Within the JAK-STAT signaling pathway, ESM-iBET specifically downregulated STAT5A gene expression, in line with earlier reports [60,61,62], while upregulating SOCS6 and PIAS3, both of which are negative regulators of phosphorylated JAKs that act to dampen JAK-STAT signaling [47]. Additionally, we demonstrate a downregulation of RELA, CASP1, CASP4, and CASP5 gene expression by ESM-iBET treatment, which are key functional effectors downstream of the canonical NF-kB and NOD2 signaling pathways, respectively. 

Overall, the monocyte transcriptomic analysis demonstrates a potent effect on multiple pathways of potential therapeutic relevance to CD by ESM-iBET. Current biologic therapies are aimed at targeting specific cytokines or pathway components, which can only be beneficial to patients in which this particular pathway is predominantly driving inflammation. The advantage of a CES1-targeted iBET (ESM-iBET) is that it can interfere with multiple CD-relevant inflammatory pathways simultaneously in monocyte/myeloid cells expressing CES1, while minimizing broad iBET effects in non-CES1 expressing cells. Whether specific targeting of mononuclear myeloid cells would demonstrate clinical benefits remains uncertain. In a complex in vivo environment, other cell types, such as intestinal B cells, T cells, epithelial, and stromal cells, all contribute to intestinal inflammation. 

## 5. Conclusions

We demonstrated a specific pattern of CES1 expression in CD patients, which is confined to monocytes, macrophages, and DC populations, across blood and local inflamed tissues of CD patients. We demonstrated the increased potency of ESM-iBET (GSK3361191) in CES1-expressing monocytes compared to the non-targeted control, iBET (GSK3235220). We also defined ESM-iBET targets at the transcriptional level in the peripheral monocytes, which are therapeutically relevant in CD patients. 

## Figures and Tables

**Figure 1 cells-11-02846-f001:**
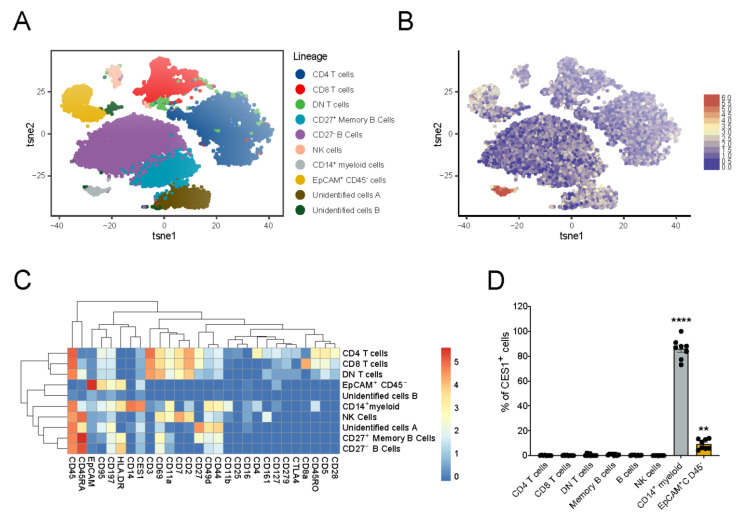
**Immunophenotyping of IBD intestinal biopsies reveals specific CES1 expression in CD14^+^ myeloid cells population.** (**A**) Intestinal biopsies were collected from 6 IBD patients. In total, 8 biopsies were collected (non-inflamed biopsies, n = 2, inflamed biopsies, n = 6, two pairs of non-inflamed and inflamed biopsies from the same patients, n = 2). Cells from intestinal biopsies of 6 IBD patients were barcoded, stained, and pooled for CyTOF and visualized in a tSNE plot after mass cytometry analysis. B cells, CD4 T cells, CD8 T cells, DN T cells, CD14^+^ myeloid cells, EpCAM^+^ cells, and NK cells were identified. (**B**) CES1 expression intensity was demonstrated among identified cell clusters in (**A**). (**C**) A heat map shows expression intensity of different (lineage) markers in relation to the identified cell clusters in (**A**). (**D**) Percentages of CES1^+^ cells among identified cell subsets are shown. Data are represented as mean with SEM of 8 samples; CD14^+^ myeloid cells, and EpCAM^+^ CD45^−^ cells were compared to the rest, Statistical testing was performed using a one-way ANOVA test; ** *p* ≤ 0.01, **** *p* ≤ 0.0001; SEM, standard error of the mean.

**Figure 2 cells-11-02846-f002:**
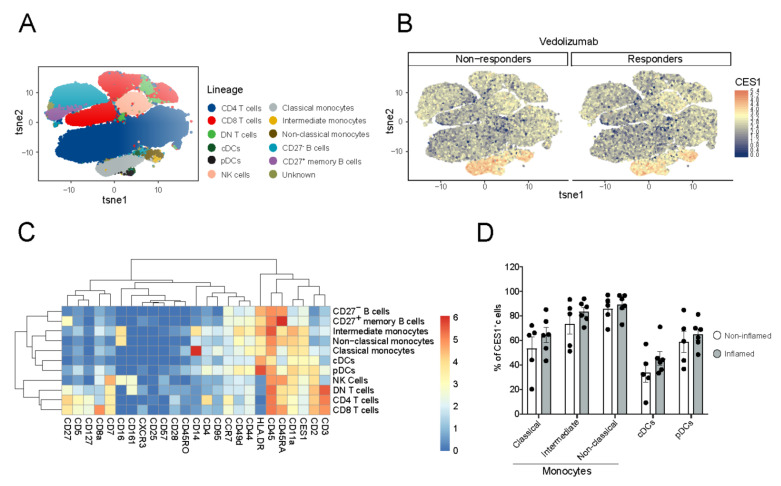
**CES1 is expressed in the mononuclear myeloid cells of peripheral blood from CD patients independent of therapy response.** (**A**) PBMCs from 11 CD patients (vedolizumab responders; n = 5, and vedolizumab non-responders; n = 6) were barcoded, stained, and pooled for CyTOF and visualized in a tSNE plot after mass cytometry analysis. B cells, CD4 T cells, CD8 T cells, DN T cells, NK cells, classical monocytes, intermediate monocytes, non-classical monocytes, cDCs, and pDCs were identified. (**B**) CES1 expression intensity in PBMCs was demonstrated among identified cell clusters in (**A**); responders (non-inflamed) and non-responders (inflamed) are displayed separately. (**C**) A heat map showing expression intensity of different (lineage) markers in relation to the identified cell clusters in (**A**). (**D**) Percentages of CES1^+^ cells are shown among identified mononuclear myeloid cells subsets (pDCs, classical monocytes, cDCs, intermediate monocytes, non-classical monocytes); responders (non-inflamed) and non-responders (inflamed) are compared.

**Figure 3 cells-11-02846-f003:**
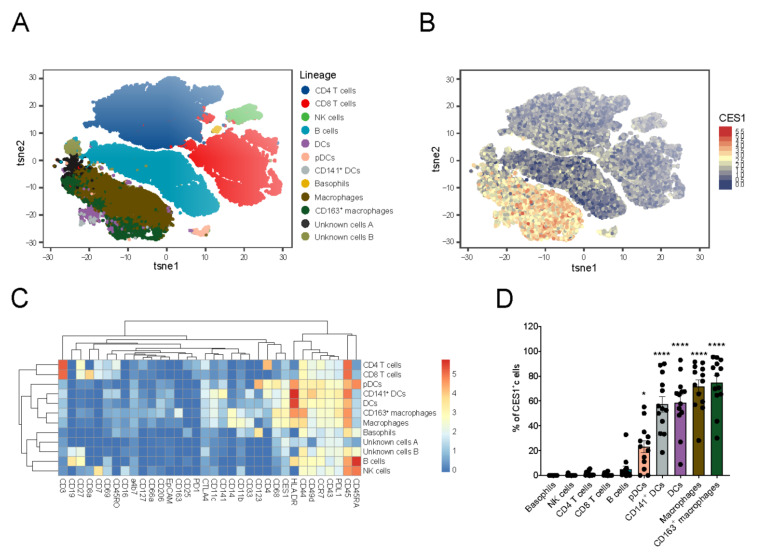
**CES1 is expressed within macrophages and dendritic cells retrieved from inflamed fistula tracts of CD patients.** (**A**) Fistula cells from CD patients (n = 13) fistula tract scrapings were barcoded, stained, and pooled for CyTOF and then visualized in a tSNE plot after mass cytometry analysis. Basophils, B cells, CD141^+^ DC type1, CD4 T cells, CD8 T cells, NK cells, macrophages, CD163^+^ resident macrophages, pDCs, and DCs were identified. (**B**) CES1 expression was demonstrated among identified cell clusters in (**A**). (**C**) A heat map showing expression intensity of different (lineage) markers in relation with the identified cell clusters in (**A**). (**D**) Percentages of CES1^+^ cells are shown among identified cell clusters in (**A**). Data are represented as mean with SEM of 13 patients; pDCs, CD141^+^ DCs, DCs, macrophages, and CD163^+^ macrophages were compared to the other cells. Statistical testing was performed using a one-way ANOVA test; * *p* ≤ 0.05, **** *p* ≤ 0.0001. SEM; standard error of the mean.

**Figure 4 cells-11-02846-f004:**
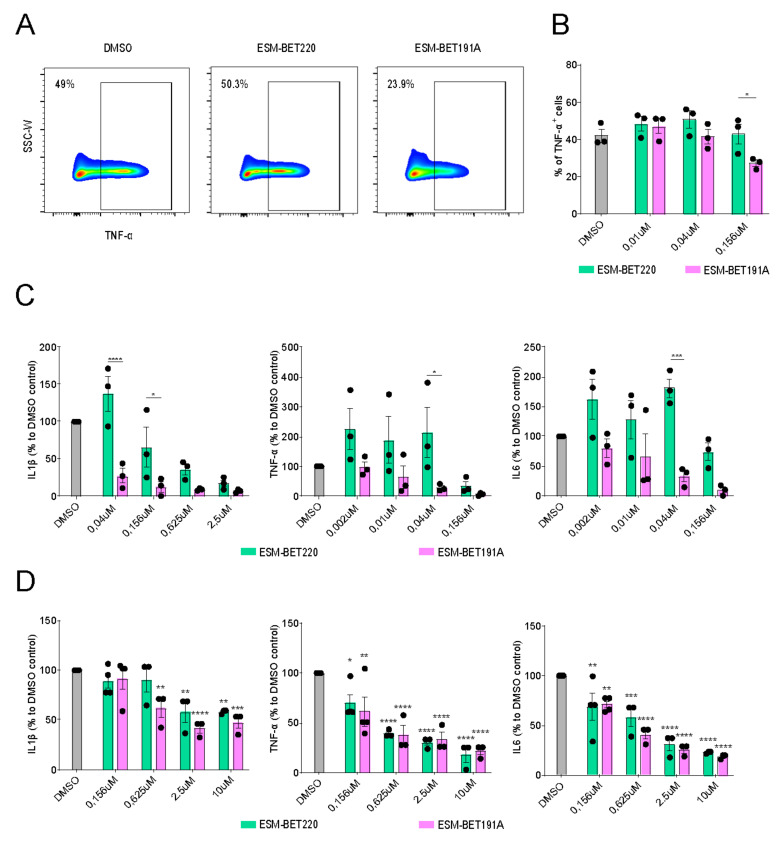
**The capacity of ESM-iBET and its non-hydrolysable control iBET to inhibit cytokines production in healthy donor CD14^+^ monocytes and CD fistula tract-derived immune cells.** All cell cultures were performed in a concentration range of 0.002–10 µM, and we visualized the concentrations, which demonstrated a clear difference between the iBET and the ESM-iBET (**A**) Representative flow cytometry plots showing TNF expression among CD14^+^ monocytes in PBMCs pre-treated with DMSO, iBET GSK33611910, or ESM-iBET GSK3235220 for 1 h, followed by LPS 40 ng/mL stimulation overnight. Representative plots are shown and are pre-gated on CD14^+^ cells after the exclusion of dead cells, CD3^+^, and CD19^+^ cells. (**B**) The percentage of TNF^+^ cells among total CD14^+^ monocytes in PBMCs is shown (n = 3). (**C**) IL1β, IL6 and TNFα protein secretion are measured with CBA in the supernatant of CD14^+^ isolated monocytes from healthy donors (n = 3), pre-treated with DMSO, iBET GSK33611910, or ESM-iBET GSK3235220, followed by LPS 10 ng/mL stimulation overnight. (**D**) IL1β, IL6, and TNFα protein secretion are measured with CBA in the supernatant of ex vivo CD fistula tract-derived immune cells, pretreated with DMSO, iBET GSK33611910, or ESM-iBET GSK3235220 (n = 3–4). Data are represented as mean with SEM of three to four donors/patients. In (**B**,**C**), similar doses of iBET GSK33611910 or ESM-iBET GSK3235220 treatment are compared. In (**C**,**D**), iBET GSK33611910 or ESM-iBET GSK3235220 are compared to the DMSO control. Statistical testing was performed using a two-way ANOVA test; * *p* ≤ 0.05, ** *p* ≤ 0.01, *** *p* ≤ 0.001, **** *p* ≤ 0.0001. SEM; standard error of the mean.

**Figure 5 cells-11-02846-f005:**
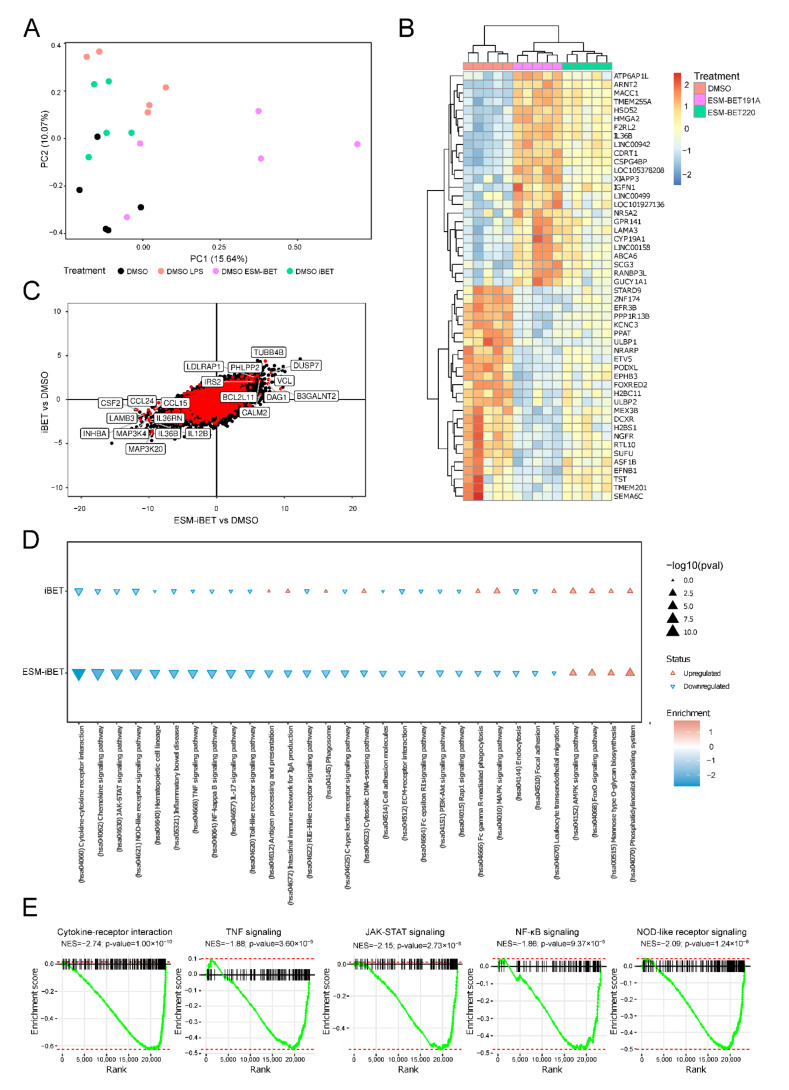
**ESM-iBET influences the transcription of inflammatory related genes and pathways with increased potency over iBET in blood CD14^+^ monocytes.** (**A**) RNA sequencing data of peripheral CD14^+^ monocytes (n= 5 healthy donors) pre-treated for 1 h with DMSO, 40 nM ESM-iBET (GSK33611910), or 40 nM iBET (GSK3235220), then stimulated with LPS (4 h). Principle component analysis is shown. (**B**) A heat map depicting the top 20 up- and downregulated genes when comparing BET inhibitors vs. DMSO. Colors represent the scaled log2 (counts). (**C**) Comparison of the Wald statistic obtained from DESeq2 when comparing ESM-iBET with DMSO, and iBET with DMSO on the x- and y-axes, respectively. In red are genes that encode proteins involved in the inflammation-related pathways; highlighted genes are the top 10 up- and downregulated genes (DEGs), comparing ESM-iBET (n = 5) vs. DMSO (n = 5) pre-treated, LPS-stimulated monocytes. (**D**) Gene set enrichment analysis (GSEA) of inflammation-related pathways with functional annotation using Kyoto Encyclopedia of Genes and Genomes (KEGG) pathways, the direction of the arrow indicates either up- or downregulation, while the size and shading of the arrow represent the –log10 (*p*-value) and normalized enrichment score, respectively. (**E**) Enrichment scores of the cytokine–cytokine receptor interaction, TNF signaling, JAK-STAT signaling, NF-kB signaling, and NOD-like receptor signaling pathways in the CD14^+^ monocytes (n = 5 healthy donors), pre-treated for 1 h with DMSO, ESM-iBET (GSK33611910), or iBET (GSK3235220), then stimulated with LPS (4 h).

## Data Availability

The datasets analyzed during the current study are available from the corresponding author on request.

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
