# Peer review of "A BET Protein Inhibitor Targeting Mononuclear Myeloid Cells Affects Specific Inflammatory Mediators and Pathways in Crohn’s Disease"

_cells, 2022, doi:10.3390/cells11182846_

Round 1

Reviewer 1 Report

This manuscript presents interesting data on the possible application of BET inhibitor containing esterase sensitive motif (ESM-iBET)in the trestment of inflammatory bowl disease. It comprises two parts - (i) the first one aiming to specify the expression of carboxylesterase-1 (CES1) in cell populations of Crohn’s disease patients’ intestinal tissue and peripheral blood and (ii) evaluating the effect of ESM-iBET in isolated healthy donor CD14+ monocytes.

The manuscript contains large amount of scientifically sound data and is significantly contributing to the field of study. Nevertheless, I have few concerns and questions to the study, which answering might help improve the scientific value of the study.

  1.  in the first part of the study (Fig. 1) you refer to the paired samples of non-inflamed and inflamed biopsies obtained from the same patients, nevertheless the results, which might be of high value when describing the specificity of CES1 expression in the inflammatory tissue are not presented. Is there indeed any increase in the CES1 expression in the inflammatory tissue compared to the non-inflamed one? Could you include these data?
  2. Your data do not consider the distinct pathogenesis of the Crohn’s disease from the ulcerative colitis. Could the CES1 expression and absence of dependency of treatment response reflect this issue and possible different variability of the cells showing high CES1 expression between these two types of IBD?
  3. The CES1 expression pattern in the inflamed fistula tracts compared to the IBD biopsies would need to be discussed more also in context to the results of the second part of the study that suggest the beneficial use of ESM-iBET. Which cells/types of cells and inflamatory types would - based on your data- be the best target for the therapy? In the current state the study is not suggesting the answer.
  4. Is the LPS- induced inflammatory model the best one? Does it correspond with the sterile inflammation, which is usually the causative and accompanying mechanism of the IBD?
  5. Minor issue - check the line 377 and correct the symbols  in the units of the concentration range. 

Author Response

Please see the attachment: 'rebuttal to the reviewers, section reviewer 1'. 

General remarks:

In our rebuttal we provide the following;

  • Rebuttal letter with a thoroughly point to point replies addressing all reviewers comments. When referring to the lines in the manuscript, we conformed to the ‘simple Markup’ form.
  • Three Supporting data files that contains additional data to support our rebuttal letter.
  • A revised manuscript text in a track change mode.

For main and supplementary figures:

  • We now have 5 main figures (instead of 7) and 6 supplementary figures (instead of 4), as we combined figures 5, 6 & 7 into figure 5 and re-trafficked the rest of data into 2 extra supplementary figures (S1, S2) upon reviewer suggestion.
  • We modified Figure 1 and 3 to add statistics stars in panels 1D and 3D respectively.

Reviewer 2 Report

This manuscript presents the results of a comprehensive set of experiments done to provide in vitro data to support the use of a molecular construct as an anti-inflammatory agent in the treatment of Crohn's disease (CD). First, the authors confirmed the expression of CES1 in CD14 expressing myeloid cells. Using a complex of ESM and iBET, the authors show that this combination can be used to attenuate inflammation in monocytes from healthy subjects and cells scrapings from fistulous tracts of CD patients. The authors evaluated the transcriptomic changes that underlie the anti-inflammatory property of ESM-iBET, wherein they show differential expression of multiple genes due to ESM-iBET.

The study is interesting, and the manuscript is well written. The study attempts to answer a question with significant clinical importance.

Minor comment

Discussion – lines 425 to 439: The discussion here indicates that CES1 expression is confined to CD14 expressing myeloid cells, inflamed intestinal epithelial cells, and cell scrapings from CD fistulous tracts. In the Results, the authors state thus: “Interestingly, we demonstrated a notable CES1+ cells within the EpCAM+ fraction (epithelial cells), compared to B cell, T cell or NK cell fraction, however much less compared to CD14+ myeloid cells. The median CES1 expression did not differ between inflamed and uninflamed samples (data not shown).” This statement, however, indicates that epithelial cells also show CES1 expression with no difference between inflamed and uninflamed epithelial cells. The major thrust of this work is to develop a molecular construct that is specific for inflammatory cells and spares the gut epithelium so as prevent treatment-induced colitis. In view of this, it is important that the authors are explicit regarding the level of expression of CES1 in inflamed and non-inflamed bowel epithelial cells and quantify the levels of these expressions in relation to that seen in the different inflammatory cells.

Author Response

Please see the attachment: "rebuttal letter, reviewer 2". 

General remarks:

In our rebuttal we provide the following;

  • Rebuttal letter with a thoroughly point to point replies addressing all reviewers comments. When referring to the lines in the manuscript, we conformed to the ‘simple Markup’ form.
  • Three Supporting data files that contains additional data to support our rebuttal letter.
  • A revised manuscript text in a track change mode.

For main and supplementary figures:

  • We now have 5 main figures (instead of 7) and 6 supplementary figures (instead of 4), as we combined figures 5, 6 & 7 into figure 5 and re-trafficked the rest of data into 2 extra supplementary figures (S1, S2) upon reviewer suggestion.
  • We modified Figure 1 and 3 to add statistics stars in panels 1D and 3D respectively.

Reviewer 3 Report

Major comments:

1.   Based on the single cell mass cytometry data, this reviewer is concerned that the study might be  grossly underestimating the myeloid compartment from IBD endoscopic tissues (Figure 1), where only CD14+ myeloid cells are described. This could be due to the fact that the endoscopic biopsies were cryopreserved before analysis. Myeloid cells are susceptible to cryopreservation and, even in fresh biopsies, they are known to be difficult to extract from tissues. Given that the expression of CES1 is limited to the myeloid compartment, it is important that the single cell suspension they analyze is as representative as possible to the actual composition of the intact tissue and that all relevant myeloid subsets are present. This reviewer suggests the authors to either assess CES1 expression directly in tissue by imaging techniques or to optimize the protocol of tissue processing/digestion to more accurately represent the composition of the myeloid compartment. Since there’s no mention to how the endoscopic biopsies were processed, a more detailed description on the processing/dissociation of the tissues in the Materials and Methods section is also recommended.

2.   A big portion of this work centers around expression of CES1 in myeloid populations. Given that myeloid cells normally show more non-specific binding of antibodies, the fact that this is a novel protein target and that a 2-step protocol is used, it is desirable that the authors show examples of staining and proof of specificity for CES1 staining in the different myeloid subsets.

3.   As described in the discussion in Lines 440-444, CES1 expression is already demonstrated earlier in healthy donor blood. If not described before, it would be desirable to show CES1 expression in healthy intestinal mucosa as a comparison with IBD intestinal mucosa.

4.   The expression of CES1 was high on all characterized myeloid cell subsets, with the exception perhaps of pDCs. Since all myeloid population would be targeted alike, both inflammatory and non-inflammatory, DCs and macrophages, the final consequences of such inhibition in inflamed tissues are difficult to anticipate. Is there literature on whether BET inhibition affects DCs or CD163+ non-inflammatory macrophages?

5.   In Figure 4, the concentrations of iBET/ESM-iBET vary greatly between experiments, even for different inflammatory mediators on the same target cells (Figure 4C). How have the authors chosen these different concentration ranges? Of note, the concentration ranges given for the experiment with fistula-derived cells are different in the M&M section than in the Results section and Figure 4D.

6.   It is not clear to this reviewer why the results of inflammatory mediators’ production in Figure 4C and D are shown in relative units instead of the actual measured values. Can the authors provide the data of their concentration, without normalization to the DMSO control?

7.   Although the myeloid-targeted drug shows promise as anti-inflammatory in PBMC-derived CD14+ monocytes, the results obtained in CD fistula-derived samples are not very encouraging. First, a decrease in inflammatory mediator levels was only observed at much higher concentrations than those required for CD14+ cells (2,5 uM vs 0.04 uM, respectively). Additionally, the increased sensitivity to ESM-iBET was not observed in the tissue-derived cells, even when the authors demonstrate CES1 expression in myeloid cells from fistula tracts. These observations raise several methodological concerns, as follows:

a.       It is especially concerning that the concentrations of inhibitors are much higher for fistula-derived cells, given that the authors are aware of the toxicity of these drugs in certain cell types. Have the authors evaluated toxicity of the drugs over the complete range of concentrations (10μM – 0.01μM) on the different cell subsets/tissues? Can the authors rule out that the results in Fig 4D are not due to cellular toxicity?

b.       It is remarkable that the authors don’t digest the fistula-derived tissue in any way, but mash/flush through a cell strainer. This procedure is usually appropriate for lymphocytes but not adequate for myeloid cells. Is it possible that the amount of mononuclear myeloid cells is too low in the cultures? Have the authors evaluated the viability and composition of the fistula-derived cultured cells? Is it possible that the relevant cells are lost during the processing and/or culturing of these samples?

c.       Why did the authors apply different experimental designs when culturing CD14+ or fistula-derived cells? In one experiment they pre-incubate the CD14+ cells with the inhibitors for 1 h and then stimulate with LPS overnight, while the fistula-derived cells are directly cultured with the inhibitors for 16 hours, without stimulation. What is the rationale for the changes in protocol in the experiments with fistula-derived material?

d.       Can the authors provide an explanation for the lack of increased sensitivity to ESM-iBET in fistula-derived cultures?

8.   Figures 5-7 give an extensive description of inflammatory genes and pathways are influenced by ESM-iBET in blood CD14+ monocytes of healthy donors. Although informative, the amount of data shown from the experiment in healthy-donor derived myeloid cells seems excessive to this reviewer. In this study, the authors set out to investigate the efficacy of ESM-iBET in Crohn’s disease. Could the data in Fig 5-7 summarized in one figure?

9.   Finally, and returning to point 7, it seems crucial to this study to demonstrate that ESM-iBET is effective as anti-inflammatory drug in inflammation-reprogrammed myeloid cells from CD patients. The potential relevant targets for CD described in Figure 7 are interesting only if the drug is shown to be effective at suppressing CD-associated inflammation. Can the authors provide data to substantiate their initial hypothesis on fistula-derived myeloid cells?

Minor points:

Line 74- “Multiple adverse events (AE) are reported…”  should be changed to “Multiple adverse events (AE) were reported…”.

Lines 76-77- “Redirecting iBET to specific cell types may limit the wide-range toxicity and allows efficacy at very low dose” should be changed to “Redirecting iBET to specific cell types may limit the wide-range toxicity and allow efficacy at lower dose”. 

Line 107- The authors should specify how many samples from CD and from UC patients they included, and from which segment of the intestine?

Line 180- The title after 3.1 is missing

Line 189- The authors identify B cells as CD45+CD45RA+HLA-DR+CD69+CD44+, while they include a metal-conjugated CD19 antibody in Supplementary Table A4. Did they use CD19 to identify B cells? If so, they should adjust the phenotypic description in line 189. If not, could they provide an explanation?

Line 211- Similar to the issue with B cells in line 189, pDCs are described as CD16-CD14-HLADR+++CD45RA+ CD2++. However, pDCs are generally identified by expression of CD123 and no expression of HLA-DR. Can the authors elaborate on this issue?

Lines 215-217- When describing the results in Figure 2, the authors state: “PBMCs collected from vedolizumab non-responsive patients (n=6) showed a trend towards higher percentage of CES1 positive cells in these myeloid subsets (Fig 2D), however this is not statistically significant“. As the authors themselves acknowledge, there is no statistically significant difference in expression between responders and non-responders. This sentence should be rephrased to more accurately reflect the data. Besides this, how is CES1 expression in the other lineages? Do the authors also see some expression in B cells, as identified in the fistula scrapings?

Lines 226-227- In the fistula tracts, the authors describe macrophages (CD68+CD14+HLA-DR+CD44+CD11b++) and resident macrophages (CD68+CD14+HLADR++CD44++CD11b+ CD163+). The only apparent difference between the subsets is the expression of CD163, which is more frequently associated to type 2 anti-inflammatory macrophages. Are the authors suggesting that the first macrophage subset is not resident but coming from circulation? In that case, both subsets express CD14 and could come from CD14+ monocytes. Could the authors elaborate more on this point?

Line 243- There is a hyphen missing in “TNFα expressing CD14+ monocytes”.

Line 248- Can the authors state the method by which the cytokines were quantified?
Lines 536 -550- Please specify: how many cells were isolated and stained from intestinal biopsies, fistula scraping, and PBMC samples? How many CD45+ immune cells were present after sequential CD45+ gating?
Line 574- The reference is missing in ‘CD45+ live cells were selected through sequential gating as described before’. Also, since the authors also describe CD45-EpCAM+ cells in the tSNE analysis, the description of the gating workflow should be adjusted to include them.
Line 687- Supplementary Table A4 would benefit from the inclusion of the clones selected for each antibody.

Author Response

Please see the attachment: "rebuttal letter, reviewer 3". 

General remarks:

In our rebuttal we provide the following;

  • Rebuttal letter with a thoroughly point to point replies addressing all reviewers comments. When referring to the lines in the manuscript, we conformed to the ‘simple Markup’ form.
  • Three Supporting data files that contains additional data to support our rebuttal letter.
  • A revised manuscript text in a track change mode.

For main and supplementary figures:

  • We now have 5 main figures (instead of 7) and 6 supplementary figures (instead of 4), as we combined figures 5, 6 & 7 into figure 5 and re-trafficked the rest of data into 2 extra supplementary figures (S1, S2) upon reviewer suggestion.
  • We modified Figure 1 and 3 to add statistics stars in panels 1D and 3D respectively.
